# Highly Sensitive Self-Powered Biomedical Applications Using Triboelectric Nanogenerator

**DOI:** 10.3390/mi13122065

**Published:** 2022-11-25

**Authors:** Tapas Kamilya, Jinhyoung Park

**Affiliations:** School of Mechatronics Engineering, Korea University of Technology and Education, Cheonan-si 1600, Republic of Korea

**Keywords:** triboelectric nanogenerator, self-powered sensor, biomedical application

## Abstract

The triboelectric nanogenerator (TENG) is a promising research topic for the conversion of mechanical to electrical energy and its application in different fields. Among the various applications, self-powered bio-medical sensing application has become popular. The selection of a wide variety of materials and the simple design of devices has made it attractive for the applications of real-time self-powered healthcare sensing systems. Human activity is the source of mechanical energy which gets converted to electrical energy by TENG fitted to different body parts for the powering up of the biomedical sensing and detection systems. Among the various techniques, wearable sensing systems developed by TENG have shown their merit in the application of healthcare sensing and detection systems. Some key studies on wearable self-powered biomedical sensing systems based on TENG which have been carried out in the last seven years are summarized here. Furthermore, the key features responsible for the highly sensitive output of the self-powered sensors have been briefed. On the other hand, the challenges that need to be addressed for the commercialization of TENG-based biomedical sensors have been raised in order to develop versatile sensitive sensors, user-friendly devices, and to ensure the stability of the device over changing environments.

## 1. Introduction

The evolution of technology continues to miniaturize electronic gadgets from the time of the invention of semiconductor technology [1]. The electronic gadgets used daily are mostly battery-dependent for their power requirements [2,3]. The drawback of batteries comes under consideration of their limited lifetime, and they becomes environmental hazards after their expiry [4,5,6]. Furthermore, the inclusion of DC circuitry in a device makes it heavy and expensive [7,8]. The research on renewable energy technology continues to give promising alternative to the use of conventional energy-based electronic equipment [9,10,11]. Among the alternative energy sources, the solar cell possesses some advantages over others for alternative energy generation, and a lot of research is investigating this solution [12,13,14]. However, its main drawback is its dependency on sunlight. Due to clouds and rainy seasons, the yield of the solar cell degrades [15,16,17]. On the other hand, a lot of mechanical energy gets lost due to our daily activities, such as walking, running, vibrations, body movements, or cars passing, and by nature’s activities, such as wind flow and tides [18,19]. To convert these mechanical energies to electrical energy, a lot of conversion techniques have been established by researchers, such as electrostatic, piezoelectric, electrochemical, and magnetostrictive methods [20,21,22,23,24]. To convert mechanical energy, the piezoelectric effect has shown a great contribution, and a lot of research has been carried out and continues to be on it [25,26]. Currently, for scavenging mechanical energy, the triboelectric nanogenerator [TENG] has attracted the attention of the scientific community and plenty of research has continued into different applications for the technology, which was invented a decade ago by Z. L. Wang [27,28,29,30,31,32,33,34,35]. The main advantage of this energy scavenger is the choosing of diverse materials, since almost every material takes part in triboelectrification in contact with other materials, and they become positively and negatively charged [36,37,38,39,40,41]. The working mechanism of a triboelectric nanogenerator for the mechanical energy scavenging and its conversion to electrical energy is based on contact-electrification and the induction effect [27,28]. Until now, several reports have been published on the self-powered sensing and detection applications of TENG [27,42,43,44,45,46]. As there is no use of a battery for the power source and each material can show tribo-properties when using bio-friendly materials, people have developed a lot of self-powered bio-medical sensors using triboelectric nanogenerators [47,48,49,50]. Wearable electronics for bio-medical sensing applications using triboelectric nanogenerator is a promising field in futuristic gadgets for the healthcare sector [51,52,53,54,55,56,57,58]. There are two categories of TENG-based sensors or detectors used for healthcare monitoring; one is wearable devices, and another is implantable devices. Wearable TENGs are attached to different parts of the human body to obtain a source of mechanical energy from various kind of movements, such as stretching, squeezing, running and, simultaneously, they give the corresponding physiological signals [42,44,52,53]. On the other hand, bio-friendly and biocompatible materials are used to develop implantable devices to implant in human organ, such as the stomach or muscle to obtain the corresponding physiological signal of the body organs [49,59,60]. As TENG converts mechanical to electrical energy and human body movements can be a mechanical energy source, by using TENG fitted to human body we can obtain electrical energy and, simultaneously, the device can act in self-powered sensing applications. The biggest advantage of using TENG-based sensing systems for bio-medical application is that there is no use of an external power source; additionally, they are lightweight, have simple features, and are low-cost. However, there are some challenges, such as sensitivity for very weak forces, and stability of performance over various ambient conditions.

In this review, we have presented a brief description about the fundamentals of TENG and a comprehensive overview of the wearable electronics and human machine interface for bio-medical sensing applications using triboelectric nanogenerator over the last 7 years. We will discuss the bio-medical sensing applications for the various body motions, monitoring heartbeats, detection of physiological signals, and respiration, vibration, and tactile sensing. The factors influential to the sensitivity of the bio-medical sensors has been discussed. In every section we have presented the materials, structures, working principle, and performance of the applications according to the corresponding cited work in the literature. Finally, the potential development and challenges of implementation of wearable TENGs in future have been discussed.

## 2. Fundamentals of TENG

Triboelectrification is a natural phenomenon observed in our daily life, and it occurs when two different materials come in contact with each other. Initially, it was considered as one of the adverse phenomena to the industries, until the first useful implementation of the triboelectric effect was developed by Van de Graaff through the famous Van de Graaff generator [61]. Later on, a triboelectric series was established by Alpha lab in 2009, whereby they have shown the triboelectrification of different materials and created a table with the polarity of them [62,63,64]. On the basis of electron affinity and, hence, the ability of attraction or repulsion of electrons by a material while it is brought under contact to other materials, it is classified as a positive and negative tribo-material, as shown in Figure 1a; human hair, skin, and nylon are positive, while cellulose, PVC, and Teflon are negative tribo-materials [62]. It will be a highly efficient triboelectric nanogenerator if two materials are chosen from the top opposing ends of the triboelectric series because, after friction, the produced surface charge density will be higher. There are several factors affecting the output performance of a TENG other than surface charge density. Figure 1b shows that the factors responsible for the output of TENG come under the surface properties of materials and environmental effects. The influencing factors, such as frequency and force, and the environmental factors, such as humidity, temperature, and presence of gaseous molecules, are also the considerable parameters which decide the output. In 2015, Wang, et. al established a formula of the figure of merits for TENG, which is summarized in Figure 1c [35,64]. It is shown that the performance figure of merit (FOM_P_) for TENG consists of a material FOM (FOM_M_) and structural FOM (FOM_S_). The FOM_P_ is considered as a standard equation to determine the performance of TENG. The most important material-dependent parameter is surface charge density, although there are some external factors which restrict the ability to generate surface charge [64].

The fundamental principle of TENG was first established by Z. L. Wang to scavenge mechanical energy, and his group showed for the first time the utilization of triboelectric effects for industry use [27,65]. The working principle of TENG is a coupling effect of contact electrification and electrostatic induction [27,66]. Here, when a material comes in contact with another material, the surface charge transfer takes place between the materials. Depending on the transfer of electrons from one material to another, one material becomes positively charged and another becomes negatively charged and, accordingly, a triboelectric series is established [64]. Due to the electron induction phenomena, an opposite kind of surface charge is induced on the electrodes attached to the materials and, hence, an electric potential develops in-between the two electrodes attached to the materials [27,64]. This developed electric potential in-between the two electrodes changes its polarity with the contact separation process and delivers alternating current at the output load. The two tribo-layers are separated by a gap, and this can be considered as a parallel plate capacitor [67,68]. The fundamentals of the presented TENG are based on Maxwell’s displacement current [66]. There are four fundamental modes of operation of TENG, as follows.

### 2.1. Vertical Contact-Separation (C-S) Mode

This mode is the widely applied mode for developing TENG configurations, as it is easy to fabricate and straightforward, as shown in Figure 2a. The working principle is as follows: when two different materials with attached electrodes comes into contact with each other under the presence of external force, they produce a surface charge depending on their electron affinity. When the two charged surface starts to separate, an electrical potential develops, which induces the opposite kind of charges on the surface of the electrodes. This induction of charges comes from the transfer of electrons from one electrode to the other electrode. During the approach, the potential difference tries to minimize and the direction of flow of the electrons becomes opposite to the case of the separation. Hence, in this way, we obtain alternating current at the output [38,69,70].

### 2.2. Lateral Sliding (LS) Mode

As the name suggests, it generates electricity by sliding the top dielectric layer over the bottom layer, as shown in Figure 2b. The top layer is positive, and bottom later is triboelectrically negative, as per the triboelectric series. When the two layers are in full contact with zero relative displacement, they possess equal and opposite charge density and, thus, no net potential develops at the electrodes. While the top layer slides outward with respect to the bottom layer, a large number of charges becomes unpaired, and that is why a potential develops between the two electrodes and, thus, electrons flow from one electrode to other electrode to counter the potential difference. The flow of current continues until the top tribo-layer slides completely with respect to bottom layer. Again, when the top layer tends to slide inward, the direction of flow of electrons becomes opposite to the previous case. Furthermore, the flow of electrons continues until the electrostatic equilibrium is established between the materials and, hence, the electrodes. In this fashion, the alternating current we obtain at the output is generated [38,71].

### 2.3. Single Electrode (SE) Mode

Although vertical C-S mode is widely used and more effective friction takes place in the lateral slide mode, these are not suitable for developing a miniaturized device because of the large device size and relatively complex circuitry. Here, a single electrode mode comes into play to take the part of the role for a small-sized device. Here, only one electrode is needed, and the ground acts as another electrode, so the effective flow of electrons for electricity generation takes place between the electrode and the ground, as shown in Figure 2c. It is the simplest working mode of TENG which operates in both vertical C-S mode and lateral sliding mode [71].

### 2.4. Freestanding Triboelectric (FST) Layer Mode

When a triboelectric layer is movable, such as by sliding or rotating, this mode comes into play with greater efficiency than the single electrode mode. There is no need to attach an electrode and connect wires to the moving triboelectric object. In this mode, two symmetrical electrodes are placed at the same plane and parallel to each other with a small gap, as shown in Figure 2d. The size of the dielectric tribo-layer is the same as the electrode. The movement of the free tribo-layer in-between the electrodes develops potential between them and, consequently, to balance this, the potential electrons flow from one electrode to another. With the change in direction of the moving dielectric layer, the flow of electrons changes its direction and produce alternating electric output [72,73,74].

Of the self-powered systems for human health monitoring based on TENG, there are two available ways which have been developed by researchers, namely an implantable device and a wearable electronic system. The implantable TENGs are environmentally friendly and biodegradable, and are implanted into different parts of the human body, such as the muscles, heart, stomach, etc., to harvest electrical energy from their stretching, vibrations, or dynamicity; this helps to diagnose health conditions by monitoring and analyzing the electrical signals coming from the self-powered TENG [74,75,76]. On the other hand, wearable TENGs are attached to human body parts, specifically different joints, the chest, and the throat, to scavenge electrical signals from the movements and to obtain different physiological electrical signals.

Here, we will be briefing the research work carried out previously based on self-powered wearable TENG used for healthcare monitoring, which is divided into two major parts. One is breathing/respiration sensing systems, and another is tactile/human activity sensing. A schematic is shown in Figure 3 representing some versatile applications of self-powered TENGs.

## 3. TENG Based Tactile/Human Activity Sensing

### 3.1. TENG for Humidity Resistor and Gait Sensor

Wearable and flexible electronic gadgets are an emerging technology for the next-generation electronic devices to simplify the communication between the human body and multifunctional electronic devices. Sweat is one of the most popular ways to diagnose the health of the human body. In most cases, sweating occurs due to physical work by the human body and, by analyzing the presence of the components of sweat, such as the concentration of electrolytes, people can monitor and diagnose the health conditions of the human body [76]. Indeed, if the relative humidity inside shoes is detected then it will be easier to an athletic person to know when to change their shoes for better performance [54]. Therefore, in 2018, Zong-Hong Lin and his co-workers developed the wearable sweat sensor, humidity sensor, and gait sensor by varying the electrodes and positioning the TENG at suitable positions [54]. In this case they have used flexible and biocompatible chitosan–glycerol-based wearable C-TENG. When the chitosan–glycerol is used as both a triboelectric material and electrode, as shown in Figure 4a, it shows constant output voltage, which is independent of the relative humidity, as shown in Figure 4b. However, the beautiful property is observed when the chitosan–glycerol is used only as a dielectric material, i.e., when a metal film is attached with the film of chitosan–glycerol and another dielectric material PTFE is attached with a metal film, and the output performance is measured under external force and at a different relative humidity. The conductivity increases with the increase in humidity of the chitosan–glycerol film and, consequently, the surface charge density decreases, as shown in Figure 4c. Based on the two results, they developed a humidity resistor through a combination of chitosan–glycerol film and C-TENG. Here, chitosan–glycerol film acted as an electrical component in the outer circuit, as shown in Figure 4d. The schematic for real-time application is shown in Figure 4e for the humidity resistor. The output voltage is taken across the film, and its conductivity decreases with the increase in humidity. At 20% relative humidity, the output voltage was 30 V, whereas the voltage decreases to 8 V in the presence of 80% relative humidity, as shown in Figure 4f. The gait phase analysis is an important measurement wing because of its application in sports, rehabilitation, and the health diagnosis of athletic people [77,78]. It is used to characterize and monitor human locomotion, specifically in athletes [79]. To do so, the Zhonh-Hong Lin group attached the C-TENG at four positions on the foot, as shown in Figure 4g, i.e., at the toe, inner side of the forefoot, outer side of the forefoot, and heel. The different shaped foot is identified by observing the electric output of a chitosan–glycerol-based gait phase detector. The electric output under force by a normal foot is at its maximum at different positions, as shown in Figure 4h; this is because the force due to body weight is distributed almost equally in the case of a normal foot. However, in the case of pigeon-toed and splayfoot feet, the force distributes unequally and, thus, we obtain unequal voltages at different positions, as in Figure 4h. The voltage bar of Figure 4h shows the visualized data coming from the different shaped foot. This experiment showed that the C-TENG can be fabricated using textiles for a self-powered humidity resistor-to-gait sensor.

### 3.2. Transparent and Stretchable Tactile Sensing

Since the introduction of TENG, the working mechanism of which is based on the coupling effect of triboelectrification and the induction of current, a lot of work has been carried out to develop a multifunctional self-powered sensor [80,81,82]. It gives some enormous advantages for wearable electronics because of its reduced circuitry, lightweight nature, and because it is applicable for a wide variety of materials. Self-powered wearable electronics for tactile sensing is an important field of research for futuristic applications in the area of healthcare. Transparency and stretchability are two important factors for wearable electronic gadgets. There are reports based on a stretchable and transparent TENG based on hydrogel and, mostly, these are single electrode systems and, thus, less efficient than a two-electrode system. However, the aqueous electrolytic solution used for the solution hydrogel becomes dehydrated over time, which causes a lagging of ionic conductivity and, hence, decreases the overall performance of the device [83,84,85]. To overcome this disadvantage, in 2019, Zhong Lin Wang and his co-workers proposed self-powered transparent and stretchable triboelectric nanogenerator for tactile sensing using a poly(2-ac-rylamido-2-methyl-1-propanesulfonic acid) (PAMPS) ionogel-based TENG [81]. Here the ionogel acted as both the triboelectrification layer and electrode. Therefore, a less complicated structure was created, which was also highly sensitive due to the patterned PDMS surface and the good conductivity of the ionogel. The patterned PDMS is sandwiched by PAMPS ionogel and, finally, it is encapsulated using PDMS, as shown in Figure 5a. The SEM image of the patterned PDMS film is shown in Figure 5b and the molecular structure of the ionogel network is presented in Figure 5c. The miniaturized device’s working mechanism is presented in Figure 5d(i–v). It is shown that the TENG sensor is highly sensitive under very low forces, which makes it attractive for wearable sensor applications. In Figure 5e–h some real-time applications related to human motions or detection of pulse have been presented. The versatility of this TENG sensor appears when its performance was tested under the flow of air. Figure 5e shows that it delivers distinct output voltage under the blowing air. This self-powered sensor can detect the pulse beat when it is attached to the wrist of a hand. Figure 5f shows it delivers 78 beats/min corresponding current pulse, which we receive at the output. It can sense stretching/twisting related to human activity and delivers corresponding output; in each case, the nature of the output is different, as shown in Figure 5g,h. It also can detect touching by a fingertip and also the bending of a finger. These features are attractive parts of this TENG sensor for real-time healthcare applications.

### 3.3. TENG for Exercise Monitoring and Rehabilitation Therapy

There are several reports based on stretchable TENGs for scavenging electrical energy and sensing for biomedical applications [84,85,86,87]. However, the most sensible and effective stretchable TENG was presented by Zhou Li and his group in 2022, whereby they reported a multi-mode triboelectric nanogenerator for energy harvesting and a self-powered sensing using a liquid metal EGaIn (eutectic gallium–indium alloy) electrode [51]. Due to good deformability, good electrical conductivity, non-toxicity, and low viscous the msw-TENG is an attractive device in the field of triboelectric nanogenerators, specifically in the domain of healthcare applications, due to its stretchable property. The msw-TENG is based on EGaIn liquid metal and silicone (Ecoflex 00-10), where the silicone is used to provide the desired shape for the liquid metal. The fluid nature of the liquid metal is responsible for the continuity of the conductivity during stretching or deformation. A structural view and the mechanical stability of msw-TENG are presented in Figure 6a. Figure 6b shows the working mechanism under contact-separation mode, where silicone rubber acted as a dielectric layer and the liquid metal as the electrode; hence, it is seen that the device works under the mechanism of a single electrode mode TENG. The electrical output measurements are depicted in Figure 6c, and these are 39 V, 0.7 µA, and 13 nC, respectively. Figure 6d shows the versatility of the msw-TENG in different states where the device performances are not affected by its stretched or twisted condition compared to the original state. Because of the distribution of the liquid metal through the patterned shape in the silicone, the connectivity and conductivity remains intact even when the strain is around 300%. The bottom side figures of Figure 6d are the simulation of stress distribution using COMSOL software under the stretched condition of msw-TENG. The higher output is shown in Figure 6e. The output voltage of msw-TENG has been examined using different bending angles of the finger, as shown in Figure 6f. In the stretching condition, the coupling between stretched state and contact-separation process helps to generate higher surface charge density. The attractive feature of the device as an active sensor is its sensitivity up to the level whereby we can measure the radial artery pulse, Figure 6g. The msw-TENG can record the vascular movement due to its structural advantage and presence of liquid metal, the properties of which remain the same under the deformed state. In Figure 6g, we can see that it detects the real-time pulse under various physiological states, i.e., at rest condition, post-exercise, and in a break state. Furthermore, by analyzing the peaks of the peripheral artery, cardiovascular disease can be diagnosed. By varying the size and attaching the device at suitable positions it can be applicable for monitoring purposes in various sport-related fields, as shown in Figure 6h. The versatility of healthcare applications makes the device a promising one in the field of the sensing applications of TENG.

## 4. Respiration Sensing

Respiration is one of the top physiological process among the important biological activities of human body. By analyzing the rate of breathing, people can sense the other physical activities, such as abnormalities in the lungs, heart, sleeping, exercise, and the emotional states of human body [88,89,90]. Researchers have developed a lot of sensors for the early diagnosis of breathing rate abnormalities, as these sensors are highly necessary for providing proper medical assistance [91,92,93,94,95]. Among the various types of sensors, research on self-powered sensors has emerged as a topic of mass interest. Indeed, TENG has occupied this research field due to its versatility when choosing suitable materials and its reduced complexity for bio-medical applications. Therefore, a lot of research has been carried out on respiration/breathing sensors using varying tribo-materials and into the structure of the devices during the past seven years [96,97,98,99,100,101,102,103,104,105]. In the following, we will summarize some potential work on pulse sensor/respiration/breathing sensors based on triboelectric nanogenerator.

### 4.1. Self-Powered Pulse Sensor for Cardiovascular Disease

Nowadays, it is seen that cardiovascular diseases are one of the most concerning matters to the medicinal industry. Indeed, it is said that around 90% of cardiovascular diseases are preventable when considering the early detection of abnormalities [106,107]. There are some sensors available on the market for monitoring the cardiovascular signals, such as ambulatory blood pressure, the electrocardiograph, the sphygmograph, and electrophysiology [108,109]. In addition, with the existing sensors, in 2017, Z. Li, Z. L. Wang, and Y. Fan led a group which developed a self-powered flexible ultrasensitive pulse sensor (SUPS) based on TENG, which is highly sensible over 10^7^ cycles [107]. It consists of a combination of two tribo-active materials, namely Kapton and copper. On one side, the cu film on the back side of Kapton act as the electrode and, on the other side, a nanostructured cu film act as both the material for triboelectrification and the electrode. The TENG is based on a vertical contact-separation mode, where the TENG SUPS was encapsulated using an elastomer. The advantage of the device is based on the nanostructured surface of both copper and Kapton, which is highly beneficial for effective friction at the molecular level. The schematic of the SUPS is shown in Figure 7a. Kapton and cu serve as dielectric layers. The SEM image of the nanostructured cu and Kapton is shown in Figure 7b,c. In this study, they used a linear motor to apply a vertical force of 50 N. To show the effectiveness of the nanostructured film, different pairs were selected, using flat Kapton, flat Cu, n-Kapton, and n-Cu. Figure 7d,e reveals that the highest electrical output comes from the combination of a differently patterned surface of cu and Kapton. The open-circuit voltage and current for n-cu and n-kapton was 109 V and 2.73 µA, respectively. To show the real impact of the SUPS, it is fitted to the radial artery of a 24-year-old man and the electrical output is recorded. It is seen that the highest output arises from the nanostructured Kapton and copper, as is shown in Figure 7f. Therefore, it is revealed that the nanostructured surface has the highest impact when it comes to the nature of the surfaces. The detectivity of the device is shown in Figure 7g, where a 10 kHz mechanical vibration signal from a loudspeaker was detected and converted to an electrical output. A faster response time of 50 µS is seen in the stable output. The stability test was carried out using a linear motor with an applied force of 30 N, and this force is 100 times higher than pulse pressure. Under this high force, the SUPS has delivered a stable output of 20 V for 0.5 million cycles, as shown in Figure 7h. The ultrasensitive property has been shown in Figure 7i, where the low amplitude of the wing of a honeybee is recorded by SUPS. The most important application of SUPS is pulse wave monitoring, which is shown in Figure 7j, and summarized in the following sentences. It is considered that the R – R interval of ECG is perceived as the sign of heart beats, and it is used as a reference for the accuracy and reliability of the pulse sensor used to track heart beats. From Figure 7j, it is seen that the peak waves of output voltage of SUPS, PPT, and PPG are synchronous to the corresponding R waves in the ECG. Again, the R–R intervals from the ECG, P-P_SUPS,_ P-P_PPT,_ and P-P_PPG_ were recorded and, by comparing and analyzing the peaks and intervals, it is shown that that SUPS has the potential for clinical use in pulse measurement.

### 4.2. TENG for Physiological Signal Sensing

Physiological signals are of great importance when recognizing the emotional state of a human being, and this is important for the applications of safe driving, social security, and healthcare [110,111]. The emotion recognition can be evaluated through the signals from speech, facial expressions, and gestures towards others [112,113]. Indeed, the apex cardiogram, radial pulsilogram, and carotid pulsilogram carry information for health assessment and the diagnosis of diseases. To sense the physiological signals, a lot of studies have been carried out using various methods [114,115,116]. In 2016, Li Wang and his group developed an ultrasensitive triboelectric sensor (T-sensor) using the rough surface of aluminum and PET as the triboelectrification layer for a vertical contact-separation mode TENG, as shown in Figure 8c. The versatile applications have been confirmed by them to show the potential of the device, which will be summarized in follows. Figure 8a shows the sensing of the corresponding movement of human hand from the stretch state to the clench state.

Weak and strong movement creates low and high output voltage, respectively. The movements of the jaw and corresponding muscles while the eyes blink is recorded by the T-sensor; although their voltage amplitude changes, the nature of their movements is clearly distinct, as shown in Figure 8b,d. We know that respiration is one of the important signals for healthcare, and it can be detected by the T-sensor. Figure 8e reveals the breathing rate in two states of the body, with the rest state and post-exercise state marked using black and red curve, respectively. The different amplitude peaks and number of peaks carry the information about the state of the body at that time. The most distinct sensing property of the T-sensor was in apex cardiogram recording, as in Figure 8f–i. The apex cardiogram is important in order to know the cardiac performance of a human. A T-sensor was attached in different positions, such as at the aorta (AO), pulmonary artery (PA), cardiac apex (CA), and tricuspid valve (TV), to measure low-frequency signals coming from the heartbeat of a healthy and fit man, as shown in Figure 8f. The apex cardiogram recording is shown in Figure 8g, where it was seen that uniform peaks corresponded to the heartbeats of a healthy man in a resting state. The ultrasensitive property of the T-sensor was presented in Figure 8h, which is the enlarged view of single heartbeat. All the phases shown in the figure are well-matched with the typical characteristics of a commercial apex cardiogram. The device can detect different important physiological pressure signals, as shown in Figure 8i, which shows its high potential in the healthcare sector.

### 4.3. Air-Flow-Driven TENG for Self-Powered Real-Time Respiratory Monitoring

There are some established sensing systems related to blood pressure, arterial pulse pressure, and heart rate systems, but respiration sensing systems comes out as the most important and widely accepted sensing systems for personal health monitoring and management systems [113,114,115,116]. In this matter, in 2018, Xudong Wang and his co-workers proposed an air-flow-driven TENG for self-powered real-time respiratory monitoring. Here, the authors developed a flexible TENG using the nanostructured surface of PTFE film and copper for tribo-materials [95]. This was a wavy-shaped PTFE film on flat copper film in an acrylic box where one end of the PTFE film was attached to the acrylic box and the other end was free to vibrate for the effective friction, with the flat copper film attached to the acrylic plane, as shown by the schematic in Figure 9a. The porosity of nanostructured PTFE film is shown using the SEM image in Figure 9b. The friction takes place in-between the PTFE and copper under the flow of air through the acrylic box, when air-flow contact-separation occurs due to the wavy pattern; the corresponding voltage and current are 2.4 V and 1.7 µA, respectively, as shown in Figure 9c,d. The operation cycle for electrical energy generation is shown in Figure 9e, the contact-separation process is illustrated by six stages in Figure 9e(i–vi), and the corresponding generated electric potential distribution developed using COMSOL software is shown in Figure 9f. Due to its wavy structure, the gap changes position-wise and, hence, the transferred charge density varies with the wavy pattern, as shown in Figure 9g, where the iv-th state generated maximum charge density and the i-th state generated the lowest charge density. The nanostructured pattern makes it highly sensitive under the force coming from air flow by respiration and, hence, inspired the development for respiration sensor. For real-time respiration sensing, the TENG was embedded inside a commercial medical mask, as demonstrated by the inset in Figure 9h. The working function is based on the air-flow by exhalation which causes vibrations of PTFE film and, hence, gives electrical output. Here, four different breathing patterns are considered, namely the deep, shallow, slow, and rapid behavior of breathing. Different breathing rates produce unequal voltage amplitude, as shown in Figure 9h–k. Figure 9l–o is the corresponding enlarged view of each voltage peak of Figure 9h–k, respectively, where it is seen how, over time, the voltage amplitude increases to peak value and decreases to zero on the basis of the contact-separation of different portions of the wavy shaped PTFE film. This successfully demonstrates the possible commercialization of the device for the monitoring of human respiration.

There are several reports for respiratory monitoring using TENG but, in most cases, these are not suitable for versatile real-time application due to the following factors: they do not have a user-friendly structure for comfortable use, or have a complex structure [97,104]. To address these issues, in 2022, Zhong Lin Wang and his group developed a comfortable and wearable real-time respiratory monitoring system, the helical fiber strain sensor (HFSS) [104]. A helical structure made using PTFE and nylon as tribo-materials; the main advantage arises because of its helical structure, which produces electrical signal even under a very small strain. The structure is shown in Figure 10a. Three states have been shown for contact-separation under strain. The working mechanism under strain is schematically presented in Figure 10b, and the corresponding distribution of potential developed using COMSOL software is given in Figure 10c. The uses of strain material and the helical structure have given the advantage of sensitivity to the device and, hence, inspired them to develop a wearable chest strap for respiration sensing. The effectiveness of the device is illustrated in Figure 10d by the real-time sensing of the breathing process. Inspiration- and expiration-led stretching and contraction was recorded. An entire breathing cycle and the corresponding voltage signal are given in an enlarged view in Figure 10d. The strap was fitted at the upper part of the abdomen for taking an electrical signal and, for each inspiration and expiration, how different body parts behave is illustrated in Figure 10e. The efficiency of the device was compared with a commercial spirometer. The results from the spirometer (Figure 10f) and from the HFSS (Figure 10g) suggest that the trend lines are a good match for each other over a similar time scale, which suggests the potential capacity for commercialization of the HFSS device for respiration monitoring.

## 5. Conclusions and Outlook

In summary, in this review, we have outlined and presented some high-potential works on self-powered biomedical applications using TENG which focused on flexibility, responsivity, sensitivity, stability, and user-friendly nature. These works took place during the last seven years, specifically in the period lasting from 2015 to the present time. Our focus was on the self-powered wearable electronic sensor, because it can be used for daily health monitoring as well as in cases of smart clothing for patients admitted to hospital for medication. We have presented only the biomedical applications which cover human motions sensing, tactile sensing, heart rate sensing, and respiration sensing. The advancement of the TENG sensors on the basis of structures, materials, and the nature of applied forces has been depicted. We have seen TENG sensor output voltage from hundreds of volts to the millivolt level, which clearly conveys the corresponding force behind the output and, hence, shows potential as the replacement for the commercial battery-functionalized biomedical sensor.

A lot of research has been carried out and is still continuing concerning the TENG-based healthcare sensor, and some of the research has shown phenomenal commercial potential. However, there are some key points researchers need to care about. Most of the TENG devices are bulky in nature, which restricts the acceptability of device. The output performances of the TENG sensor should be tested in different environments or by changing the parameters which badly affect electrical output, because a stable output in different conditions make a device attractive for use. The materials used are mostly polymer-based, so more research needs to be carried out using bio-friendly materials, such as covalent organic frameworks, or using thin films of organic molecules for further miniaturizing the TENG sensor and to obtain ultra-sensitivity of the healthcare sensors to the level where they can compete with the existing commercial biomedical sensors.

## Figures and Tables

**Figure 1 micromachines-13-02065-f001:**
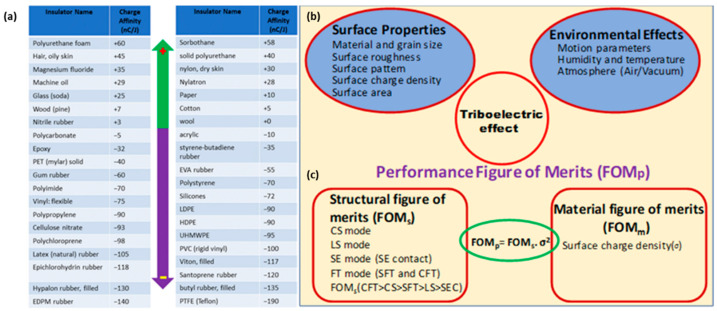
(**a**) A typical triboelectric series consists of some tribo-positive and negative materials. (**b**) A schematic representation of factors affecting the triboelectrification. (**c**) The figure of merits (FOM) for TENG. Reproduced with permission [35], copyright 2020, Elsevier.

**Figure 2 micromachines-13-02065-f002:**
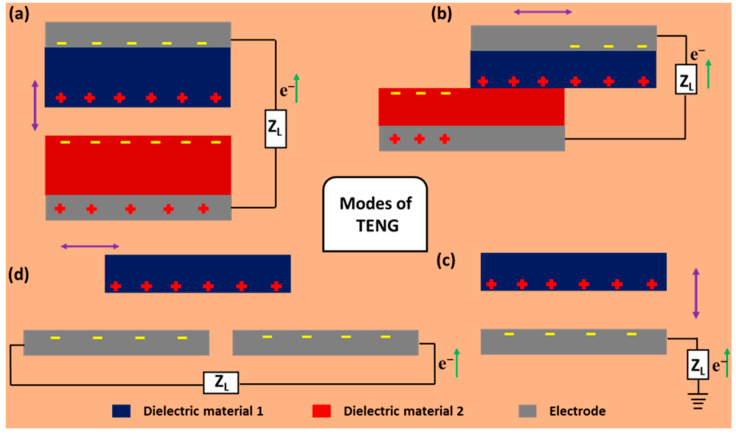
Different modes of TENG. (**a**) Vertical contact-separation (CS) mode. (**b**) Lateral sliding (LS) mode. (**c**) Single electrode (SE) mode. (**d**) Freestanding triboelectric (FST) mode.

**Figure 3 micromachines-13-02065-f003:**
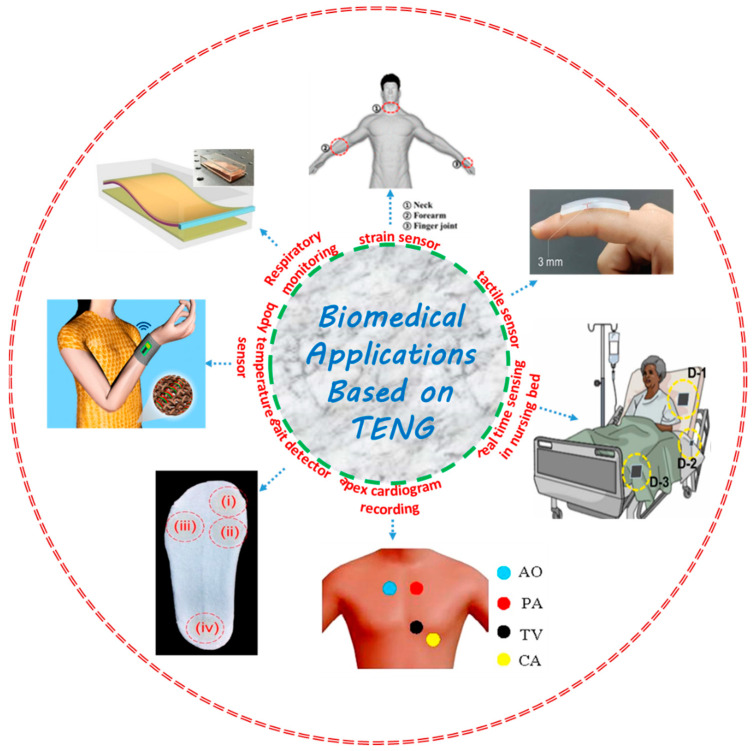
Overview of some important biomedical sensing applications using a triboelectric nanogenerator. Reproduced with permission [48] for real-time sensing in a nursing bed, copyright 2022, Elsevier. Reproduced with permission [51] for tactile sensor, copyright 2022, Elsevier. Reproduced with permission [54] for a gait detector, copyright 2018, Elsevier. Reproduced with permission [55] for strain sensor, copyright 2015, American Chemical Society. Reproduced with permission [56] for respiratory monitoring, copyright 2018, American Chemical Society Reproduced with permission [57] for apex cardiogram recording, copyright 2016, Elsevier. Reproduced with permission [58] for a body temperature sensor, copyright 2014, American Chemical Society.

**Figure 4 micromachines-13-02065-f004:**
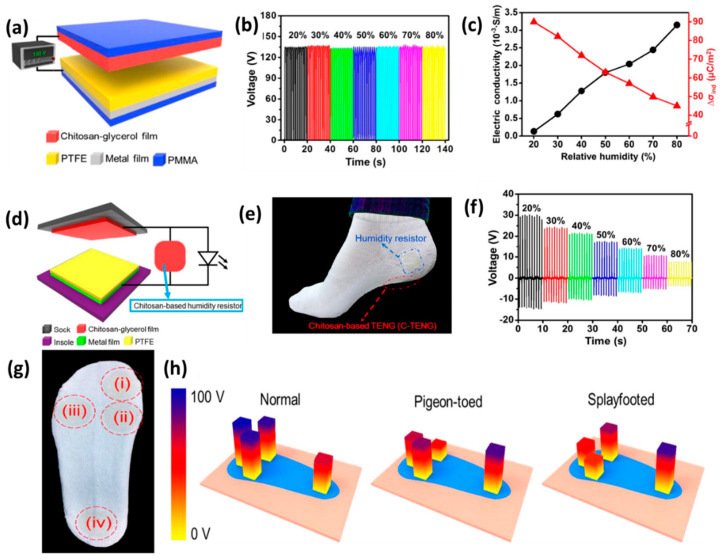
(**a**) Schematic for C−TENG. (**b**) Output voltage at different relative humidity. (**c**) The electrical conductivity and surface charge density of chitosan−glycerol film at different relative humidity. (**d**) Schematic for humidity sensor. (**e**) Photograph describing a self-powered humidity sensor combining a C−TENG and chitosan–glycerol base humidity resistor. (**f**) Electrical output voltage of a humidity resistor under the parameter of different humidity. (**g**) Photograph of gait detector whereby different positions are encircled for C−TENG fitting. (**h**) Sensing of different shaped feet. Reproduced with permission from [54], copyright 2018, Elsevier.

**Figure 5 micromachines-13-02065-f005:**
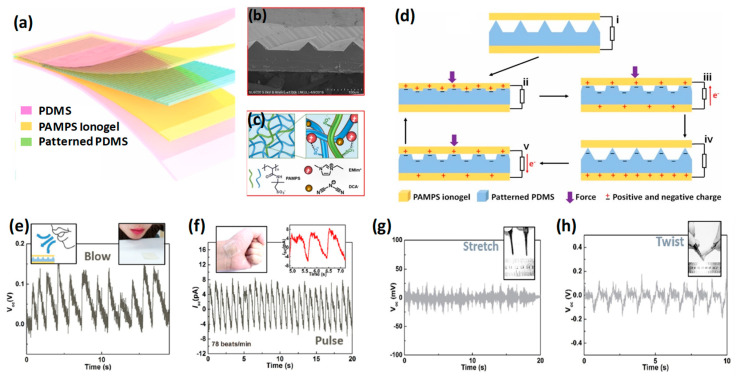
(**a**) The structure of a transparent layered TENG sensor. (**b**) The SEM image of the patterned PDMS. (**c**) Molecular structure of ionogel network. (**d**) Working principle of TENG sensor. (**e**–**h**) Different applications of the TENG sensor. Reproduced with permission from [84], copyright 2019, Elsevier.

**Figure 6 micromachines-13-02065-f006:**
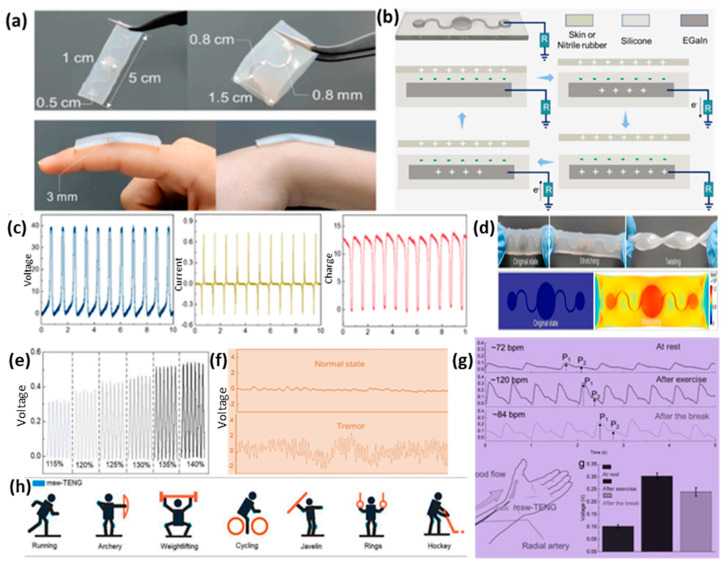
(**a**) Photographs of msw−TENG. (**b**) Working mechanism. (**c**) Output voltage, current, and generated charge by the TENG. (**d**) The simulated schematics for the stretched mode. (**e**) Output voltage at different stretched condition. (**f**) Normal and tremor state of a patient. (**g**) Output signal of the TENG for monitoring radial artery pulse signals in different motion states, showing different heart rates. (**h**) Potential application scenarios for joint monitoring based on the msw−TENG. Reproduced with permission from ref [51], copyright 2022, Elsevier.

**Figure 7 micromachines-13-02065-f007:**
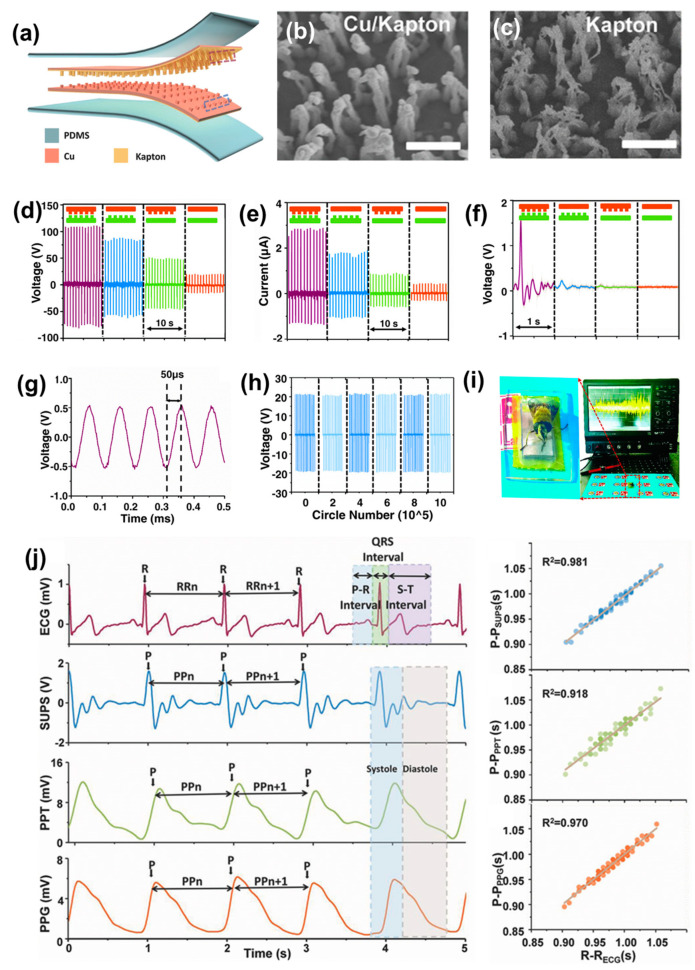
(**a**) Schematic structure of SUPS. (**b**,**c**) SEM image of nanostructured copper and Kapton surface. (**d**) Voltage output of SUPS. (**e**) Current output of SUPS. (**f**) Voltage output of a 24−year−old man when the SUPS was fitted to the radial artery. (**g**) Output voltage of SUPS under a high-frequency signal. (**h**) Stability test of the device. (**i**) Real-time optical image of bee wings and corresponding electrical output by the vibrations from the bee wings. (**j**) The pulse measurement and the linear fitting analysis of SUPS, PPT, and PPG compared with ECG. Reproduced with permission [102], copyright 2017, John Wiley and Sons.

**Figure 8 micromachines-13-02065-f008:**
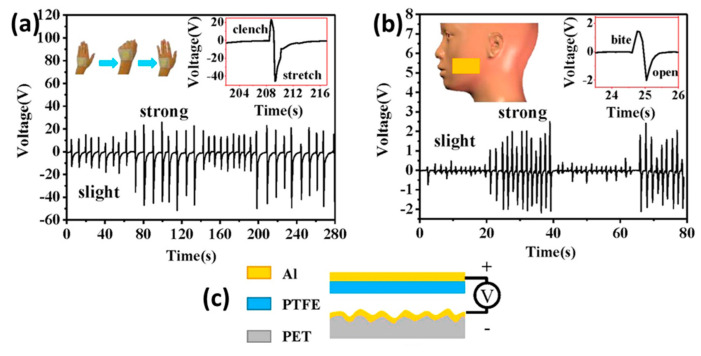
Application of T-sensor. (**a**) Detection of the clenching and stretching of hand. (**b**) The jaw movement recording. (**c**) Schematic for T−sensor. (**d**) Corresponding signal to eye’s blink. (**e**) Heart rate monitoring. (**f**) The position of the sensors mounted on different positions on the chest, such as the aorta (AO), pulmonary artery (PA), tricuspid valve (TV), and cardiac apex (CA) to record physiological signals. (**g**) Apex cardiogram (ACG) recording data of a 22−year−old man. (**h**) Detailed information of the ACG curve. (**i**) Aorta pressure curve (APC), pulmonary artery pressure curve (PAPC), and tricuspid valve pressure curve (TVPC) recorded in a healthy 22−year−old male. Reproduced with permission from [57], copyright 2016, Elsevier.

**Figure 9 micromachines-13-02065-f009:**
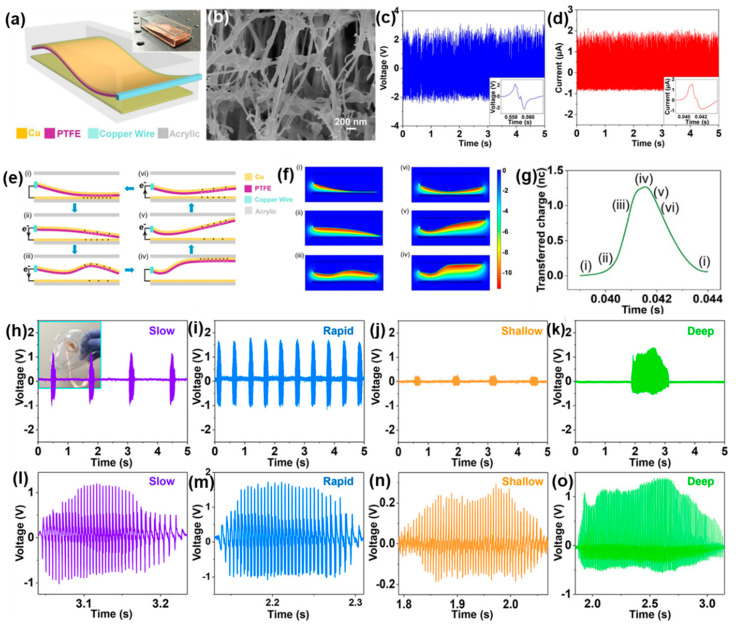
(**a**) Schematic of air-flow-driven TENG. (**b**) SEM image of the nanostructured surface of the film of PTFE. (**c**,**d**) Electrical output voltage and current at a rate of air flow of 120 L/min. (**e**) Schematic for working mechanism. (**f**) The distribution of electric potential which is simulated using COMSOL software. (**g**) Corresponding generated charge at each cycle. Real-time respiratory signal from various breathing rates. (**h**) Slow breathing rate. (**i**) Rapid breathing rate. (**j**) Shallow breathing rate. (**k**) Deep breathing rate. (**l**–**o**) Corresponding output voltage when the TENG was embedded with a medical mask. Reproduced with permission [56], copyright 2018, American Chemical Society.

**Figure 10 micromachines-13-02065-f010:**
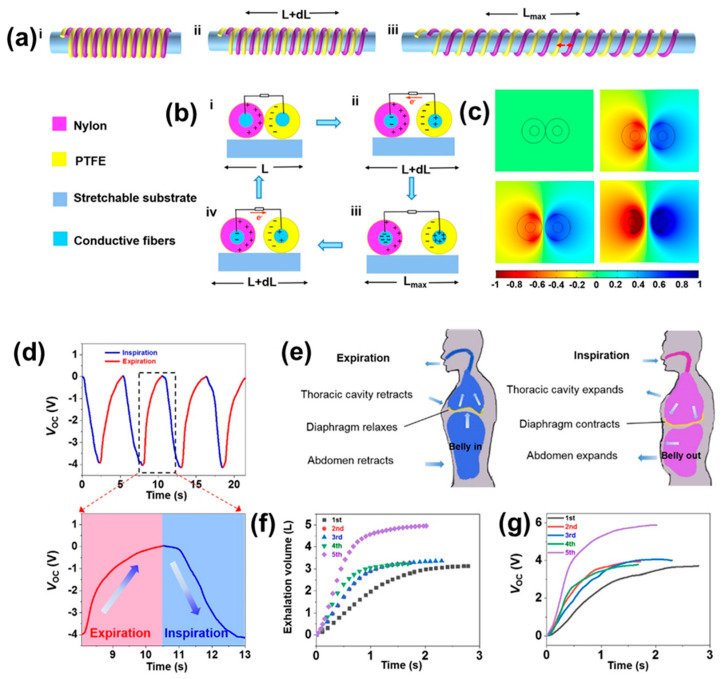
(**a**) Schematic for HFSS. (**b**) Working mechanism for the generation of electrical signals. (**c**) Potential distribution by COMSOL software. (**d**) Voltage signals for corresponding inspiration and expiration. (**e**) Movements of different body parts while breathing. Comparative performance by (**f**) commercial spirometer and (**g**) HFSS. Reproduced with permission from [104], copyright 2022, American Chemical Society.

## Data Availability

Not applicable.

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
