# Peer review of "Highly Sensitive Self-Powered Biomedical Applications Using Triboelectric Nanogenerator"

_micromachines, 2022, doi:10.3390/mi13122065_

Round 1

Reviewer 1 Report (Previous Reviewer 1)

The manuscript has been properly revised.

Reviewer 2 Report (Previous Reviewer 2)

The authors have well addressed my comments and improved the manuscript's quality after revision. So, the manuscript is accepted for publication in the present form.

This manuscript is a resubmission of an earlier submission. The following is a list of the peer review reports and author responses from that submission.

Round 1

Reviewer 1 Report

This review discusses about some fundamentals of TENG and an overview of the wearable electronics and human machine interface for sensing applications. TENG is definitely one of the most promising technologies for bio-devices, however, the connections between TENG and bio-medical sensing applications is missing in this review. Bio-motion signals are far from bio-medicals, while obviously they mixed them together. The structures and organizations are also poorly arranged, with large paragraphs from the original reports, which make it very loosely. Hence it is not recommended to be considered for publication in this journal.

1. The languages and expression need to be greatly improved for the whole manuscript. Such as: “Triboelectric nanogenerator (TENG) is a simplest electrical energy scavenging technology...”(page 1, line 7)  ; “The most important drawback of the use of piezoelectric nanogenerator is the finding out of the suitable materials because only certain materials show piezo-property.”(page 1, line 39); “Using TENG a lot of research has carried out for scavenging mechanical energy but Self-powered sensing and detection is a beautiful feather in its application part because of its less circuitry...”(page 2,line 49)....

2. In the introduction part, what is the relationship between TENG and bio-medical sensing applications? What are the benefits and challenges for this topic?

3. The authors spent large paragraphs on examples with detailed descriptions, while in a reviews, insight and comments are more important. Like in sections “ 3.2. Transparent and stretchable tactile sensing” and “3.3. TENG for exercise monitoring and rehabilitation therapy”, the whole section is almost copy word for word from a single original report. This reviewer finds it very hard to focus on such redundant descriptions. They should be more comprehensive and conclusive.

4. The authors divided the self-powered wearable TENG used for healthcare monitoring into two parts: one is breathing/respiration sensing systems and another is tactile/human activity sensing (page 5, line 169). However, in the following sections of “3. TENG based biomedical sensing systems” and “4. Respiration sensing”, they apparently differ from the division.

Reviewer 2 Report

The manuscript entitled “Highly Sensitive Self-powered Biomedical Applications using Triboelectric Nanogenerator” reviews TENG-based self-powered devices for biomedical applications. The author's attempt to review the TENG-based devices is decent and well summarized the important works. Also, the key features responsible for the highly sensitive output of the self-powered sensors have also been addressed very well. However, the reviewer has several concerns that the authors need to address before publishing. In addition, there are several mistakes present in this article, which also need to be resolved. Therefore, a minor revision is needed before publishing this manuscript in the Micromachines journal as listed below.

1.       Most importantly, I found some figures were just copied from the original review articles (E.g., Figure 1 is the replica from ref. [55]). So please make a new figure in the author's perception in order to attain better readership.

2.       In addition, the schematic diagram of different modes of TENG given in Fig.2 is wrong. There are some mistakes in the sketch. For example, the electrical connection given in Fig.2d device is wrong. It should not be connected to the ground. In this mode, the electrical connection is between two electrodes. And the symbol used for the ground is also wrong, use standard notations. So please correct these issues and sketch them again.

3.       And in the same figure, mention them as just dielectric material rather than specifying them as triboelectric positive and negative layers. Because these modes represent general device mechanisms.

4.       And the figure numbers (for example, Fig.2, Fig.4h(i-iv)), Fig.9e are wrongly mentioned in some places. So, authors should carefully check and correct them.

5.       And some of the figures are not seen properly. Please use high-resolution images and make them appear clear.

6.       In the introduction part, the explanation related to nanogenerator-based biomedical devices is very limited. Please do some more literature survey and elaborate on it.

7.       There are several typos, grammatical errors, and punctuation errors existing in the manuscript. Please correct them and refine the manuscript with the help of native English speakers.
